# Terahertz Spectroscopy: An Investigation of the Structural Dynamics of Freeze-Dried Poly Lactic-co-glycolic Acid Microspheres

**DOI:** 10.3390/pharmaceutics11060291

**Published:** 2019-06-20

**Authors:** Talia A. Shmool, Philippa J. Hooper, Gabriele S. Kaminski Schierle, Christopher F. van der Walle, J. Axel Zeitler

**Affiliations:** 1Department of Chemical Engineering and Biotechnology, University of Cambridge, Philippa Fawcett Drive, Cambridge CB3 0AS, UK; tas61@cam.ac.uk (T.A.S.); pjh200@cam.ac.uk (P.J.H.); gsk20@cam.ac.uk (G.S.K.S.); 2Biopharmaceutical Development, AstraZeneca, Granta Park, Cambridge CB21 6GH, UK; wallec@medimmune.com

**Keywords:** terahertz spectroscopy, microspheres, drug delivery, formulation development, PLGA, molecular mobility

## Abstract

Biodegradable poly lactic-co-glycolic acid (PLGA) microspheres can be used to encapsulate peptide and offer a promising drug-delivery vehicle. In this work we investigate the dynamics of PLGA microspheres prepared by freeze-drying and the molecular mobility at lower temperatures leading to the glass transition temperature, using temperature-variable terahertz time-domain spectroscopy (THz-TDS) experiments. The microspheres were prepared using a water-in-oil-in-water (w/o/w) double-emulsion technique and subsequent freeze-drying of the samples. Physical characterization was performed by morphology measurements, scanning electron microscopy, and helium pycnometry. The THz-TDS data show two distinct transition processes, $T_{g,\beta }$ in the range of 167–219 K, associated with local motions, and $T_{g,\alpha }$ in the range of 313–330 K, associated with large-scale motions, for the microspheres examined. Using Fourier transform infrared spectroscopy measurements in the mid-infrared, we were able to characterize the interactions between a model polypeptide, exendin-4, and the PLGA copolymer. We observe a relationship between the experimentally determined $T_{g,\beta }$ and $T_{g,\alpha }$ and free volume and microsphere dynamics.

## 1. Introduction

Microencapsulation has been explored as a promising method for controlled drug release [1,2,3]. Polymeric microspheres, for example polylactic acid (PLA), polyglycolic acid (PGA), and poly(d,l-lactide-co-glycolide) (PLGA), can be used as effective drug-delivery systems protecting the encapsulated active agent and controlling the release rate over periods of hours to months [1,4]. By applying the correct methodology when preparing microspheres, challenges such as targeting the drug to a specific organ or tissue, and precisely controlling the rate of drug delivery to the target site can be addressed [3,5]. The technique of double water-in-oil-in-water (w/o/w) emulsion has been widely used for the encapsulation of hydrophilic drugs as this preparative method minimizes the loss of drug activity via contact with the organic solvent [2,6]. The basis of this methodology is to emulsify an aqueous solution of the active compound in an organic solution of the hydrophobic coating polymer. Then, this primary water-in-oil (w/o) emulsion is dispersed in a second aqueous phase, forming a double water-in-oil-in-water (w/o/w) emulsion. The solid microsphere is produced as the organic solvent evaporates. In this work, we prepared blank PLGA microspheres (containing no polypeptide) of grades 50:50 and 75:25 (lactide:glycolide ratio), using the double-emulsion process [2]. Using the same technique, we also loaded PLGA microspheres at low and high concentrations of exendin-4, a glucose-dependent insulinotropic polypeptide used to treat type 2 diabetes [7]. The microspheres were subsequently freeze-dried. The objective of this work was to prepare, characterize, and understand the structural dynamics of these peptide-loaded biodegradable polymeric microspheres.

To optimize the formulation of microspheres and achieve release delivery of the active drug, the microsphere product ought to be stable [1,8]. When determining the chemical stability of a material, molecular mobility is a key factor [9,10,11] and it has been shown that an increase in molecular mobility is directly linked to an increase in the chemical degradation of a material [9] and aggregation [12], and therefore its storage stability. Thus, it is critical to understand the molecular mobility behavior of a material and its dependence on temperature, and specifically the molecular dynamics of the material leading up to the glass transition temperature ($T_{g}$). It has been shown that a great number of polymers and amorphous pharmaceuticals, ranging from small molecules to high molecular weight peptides and proteins exhibit at least two dielectric relaxation processes: the primary, or α-relaxation process and a secondary, or β-relaxation process [13,14,15]. The former can be observed at temperatures above $T_{g}$ and can be designated as $T_{g,\alpha }$, and the latter occurs at temperatures below $T_{g}$ and is associated with a secondary glass transition process, which can be indicated as $T_{g,\beta }$ [16,17,18,19,20].

In previous work we investigated these processes in neat PLGA 50:50 and 75:25 of low, medium, and high molecular weights [21]. For all the copolymer PLGA film samples we observed three regions of relaxation behavior with distinct $T_{g,\beta }$ and $T_{g,\alpha }$ transition temperatures. We showed that with an increase in temperature, the copolymer chains can transition through different conformational environments, each constrained by its characteristic potential energy landscape, and that the movement of each PLGA chain is restricted by adjacent entangled chains. At $T_{g,\beta }$, the system has sufficient thermal energy and free volume to overcome the energy barrier arising from chain entanglement, allowing for local mobility to occur. Furthermore, we show that there is a relationship between free volume and the value of $T_{g,\beta }$ for the different film samples.

Fourier transform infrared spectroscopy (FTIR) is a vibrational spectroscopy used to gain structural information about a sample. The advantage of FTIR for structural analysis is it is a non-contact technique; only a small quantity of sample is needed (∼1 mg mL${}^{-1}$); the sample can be probed in different physical environments such as the solid state, liquid state, and when adsorbed to a surface; the sample preparation is minimal and a spectrum can be obtained in a few minutes [22]. For systems with multiple components such as polymer-peptide microspheres, FTIR can be used to identify the individual components, and interactions between the components which shift the peaks [23].

Terahertz time-domain spectroscopy (THz-TDS) is a valuable technique which can be used to detect $T_{g,\alpha }$ and $T_{g,\beta }$, associated with the α- and β-relaxation processes respectively [10,13]. It is a relatively recent technique which is used to investigate the molecular dynamics of relaxation processes at high frequencies [24]. The advantage of this technique is that it is a non-contact technique and can measure molecular mobility of a material over a broad temperature range over the spectral region of 0.1–3 THz. It can be used to investigate the microscopic mechanisms of amorphous polymer dynamics [25]. In the present study we examine samples of freeze-dried microspheres with different concentrations of encapsulated peptide. We investigate the molecular mobility of these materials and the behavior and trends these exhibit with respect to the temperature dependence using THz-TDS (Table 1), and physically characterise each system (Table 2 and Table 3). Thus, the purpose of this work is to provide a comprehensive understanding of the relationship between the relaxation dynamics and the molecular structure of PLGA microspheres with varying exendin-4 peptide concentrations, and to rationalize the behavior of these materials in relation to $T_{g\alpha }$ and $T_{g\beta }$.

## 2. Materials and Methods

### 2.1. Materials

Throughout this work the polymers are referred to by their monomer ratio used. For instance, PLGA 75:25 refers to a copolymer consisting of 75% lactic acid and 25% glycolic acid. Medium-MW (10–25 kDa) PLGA 50:50 and medium-MW (20–30 kDa) PLGA 75:25 were purchased from Evonik Corporation (Birmingham, AL, USA). The exendin-4 peptide (4.2 kDa) was provided by MedImmune Limited (Cambridge, UK).

### 2.2. Microencapsulation Preparation

Blank PLGA microspheres, were prepared for the study using the water-in-oil-in-water (w/o/w) double-emulsion technique: First, 500mL of a 0.5% aqueous solution of polyvinyl alcohol (PVA) (87–90% hydrolyzed, 13–23 kDa) were emulsified in 12.5mL of a 5% (w/v) PLGA dissolved in dichloromethane (DCM) by stirring at a rotation speed of 22,000 pm for 15 seconds using an ultra-turrax (IKA T-25, Cole-Parmer, UK). This primary emulsion was emulsified into 0.5% aqueous PVA under stirring at 200rpm, on a stirring plate, for four hours. For the preparation of the peptide-loaded microspheres, exendin-4 was dissolved into a buffer consisting of citrate, citric acid, (pH = 4.5) and 0.5% PVA. This peptide solution was emulsified into the 5% (w/v) dispersion of PLGA in DCM, which was stirred at rotation speed 22,000 rpm for 15 s using the ultra-turrax. Different peptide loadings in the microspheres were achieved using exendin-4 concentrations of 1% and 10% (w/v) in the primary emulsion. Conceptually, the same method was followed to produce the peptide-loaded microspheres, as that for the blank microspheres. The solid microspheres were collected by centrifugation at 4000*g* for 5 min, then washed with distilled water three times, and centrifuged once more at 4000*g* for 5min. Finally, after removal of the aqueous supernatant and dispersion, the microspheres were freeze-dried using a lyophilizer. First, prior to lyophilization, an annealing step was performed by cooling the shelf to 233K for 240 min, raising the temperatures to 257K for 200 min and cooling the shelf again to 233K for 170 min at a pressure of 160mbar. Then, lyophilization was performed using the following steps: primary drying was completed at 233K for 30 min, and then the temperature was raised to 253K for 2440min at a pressure of 133mbar; secondary drying was subsequently performed at 313K for 960min, also at 133mbar. The vials were closed under a pressure of 266mbar at 298K using a rubber stopper, removed from the lyophilizer, and crimped with aluminum seals. Vials were stored at 278K until all further measurements and analysis. Exendin-4 concentration was determined using the bicinchoninic acid protein assay kit (MilliporeSigma, St. Louis, MO, USA) following the manufacturer’s instructions, with bovine serum albumin (BSA) used as the standard. The exendin-4 concentration was analyzed using a UV-VIS spectrophotometer (Agilent Cary 60 UV-Vis spectrophotometer, Agilent Technologies, Santa Clara, CA, USA) at 562nm. The encapsulation efficiency-based BCA assay values were determined on the liquid samples. Based on these measurements, we calculated the encapsulation efficiency to be: 15.4% and 37.12% for the low and high peptide loading of PLGA 75:25 microspheres, respectively, and 42.88% and 36.72% for the low and high peptide loading of PLGA 50:50 microspheres, respectively. This experiment was completed one time. The water content for each lyophilized microsphere was determined using Karl Fischer (Mettler Toledo, Leicester, UK) coulometric titration.

### 2.3. Helium Pycnometry Measurement

The lyophilized microsphere samples were analyzed as powders using a helium pycnometer (Micromeritics Accupyc II 1340, Norcross, GA, USA) to determine the density and specific volume of the microparticles. Approximately 20mg of each sample were placed in the instrument compartment and measurements were performed with helium gas at 298K at a pressure of 10mbar. No pretreatment conditions were required. The measurements were repeated for each sample five times. The average volume of the five repeat measurements was used to determine the density of each material. For comparison, polymer films of PLGA 50:50 and PLGA 75:25 were prepared using the vacuum compression molding (VCM) tool (MeltPrep, Graz, Austria), and these were also analyzed using helium pycnometry.

### 2.4. Morphology Measurement

The particle size and shape of the microsphere particles were characterized, for unlyophilized liquid samples and for lyophilized powder samples, using a Morphologi G3 instrument (Malvern Panalytical Ltd., Malvern, UK). Each unlyophilized liquid sample was dispersed in 1mL of 5% aqueous PVA solution to allow for spatial separation of the particles and reduction of agglomerates. The lyophilized powder samples were prepared for measurement using a dry powder disperser. The instrument captured images of the individual particles and the morphological properties for each particle were determined from the images by image analysis using the Morphologi G3 software (Malvern Panalytical Ltd., Malvern, UK).

### 2.5. Scanning Electron Microscopy

The range of morphologies of the microspheres were characterized using scanning electron microscopy (SEM), with a Zeiss CrossBeam 540 instrument, equipped with a Gemini 2 column (Carl Zeiss Microscopy GmbH, Jena, Germany). To qualitatively analyze the shape and surface of the samples, the lyophilized microspheres were sputter coated with gold using an Emitech 550 (Emitech Ltd., Ashford, UK). The samples were then examined under vacuum at an acceleration voltage of 1kV, and imaged using secondary electrons via an Everhart-Thornley detector (Carl Zeiss SMT GmbH, Oberkochen, Germany).

### 2.6. Differential Scanning Calorimetry (DSC)

A Q2000 Differential Scanning Calorimeter (TA Instruments, New Castle, DE, USA) was used to determine the calorimetric glass transition temperature ($T_{g,\mathrm{DSC}}$, defined by the onset temperature) for each material. 2–3 mg of sample material were placed in hermetically sealed aluminum pans under a constant flow of nitrogen atmosphere (flow rate of $50\;\mathrm{mL}\;\min^{-1}$) and heated at a rate of 10Kmin${}^{-1}$ to 358K, and subsequently cooled down to 293K at 40Kmin${}^{-1}$. Finally, the samples were heated from 293K through $T_{g}$ to 358K again at a rate of 10Kmin${}^{-1}$. The temperature and heat flow of the instrument were calibrated using indium ($T_{m}=430\;$K, $\Delta H_{\mathrm{fus}}=29\;J\;g^{-1}$).

### 2.7. Fourier Transform Infrared Spectroscopy

FTIR was used to examine the change in the secondary structure of the PLGA 75:25 microspheres at 278K. For FTIR analysis, each microsphere material (300μg) was mixed with 100mg potassium bromide (KBr) using an agate mortar and pressed into 7mm self-supporting disks using a load of 10Tons. FTIR spectra were acquired using a Cary 680 FTIR spectrometer (Agilent Technologies, Santa Clara, CA, USA) with 60 scans and a resolution of 1cm${}^{-1}$. At least four spectra were measured for each material. The recorded spectra were normalized based on the total area under the curve [26].

### 2.8. Terahertz Time-Domain Spectroscopy (THz-TDS)

#### 2.8.1. Sample Preparation and Experimental Methodology

For each sample 70mg were weighed in under atmospheric protection in a glove bag (AtmosBag, Merck UK, Gillingham, Dorset, UK) which was purged with dry nitrogen gas (relative humidity <1%) to avoid moisture sorption from atmospheric water vapor during preparation. The lyophilized powder samples were pressed into 13mm diameter disks using a load of 1.5tons. The tablets were between 300–650 μm in thickness each, and were placed between two z-cut quartz windows of 2.05mm thickness. This sandwich structure was sealed in the sample holder, and used immediately following preparation for THz-TDS measurements.

The THz-TDS spectra were acquired using a commercial TeraPulse 4000 instrument across the spectral range of 0.2–2.2 THz (TeraView, Cambridge, UK). The sample temperature (90–360 K) was controlled using a continuous flow cryostat with liquid nitrogen as the cryogen (Janis ST-100, Wilmington, MA, USA) as outlined previously [27]. The cryostat cold finger accommodated both the reference (two z-cut quartz windows) as well as the sample (quartz/sample/quartz sandwich structure as described above). The two z-cut quartz windows that were used for the reference (same thickness and diameter dimensions as sample) were directly pressed to one another without any spacer in between the two windows to avoid internal reflections in the time-domain signal. The cryostat cold finger was moved vertically using a motorized linear stage to switch between sample and reference at each measurement temperature. For each temperature, the sample and the reference were measured at the center position, with 1000 waveforms co-averaged for each acquisition, resulting in a measurement time of approximately 1min for each sample.

The temperature of the sample was measured using a silicon diode mounted to the copper cold finger of the cryostat. The temperature controller used was a Lake Shore model 331 (Westerville, OH, USA). For each series of measurements, a sample and a reference were loaded into the cryostat, the cryostat chamber was evacuated to 10mbar and the cold finger was cooled to a temperature of 90K. The cryostat was allowed to equilibrate for 10–15 min at 90K and the first set of sample and reference measurements was acquired. Subsequently the cold finger was heated using temperature intervals of 10K (at a rate of 2Kmin${}^{-1}$). At each desired temperature point the system was allowed to equilibrate for 3min before a set of sample and reference measurements were acquired.

#### 2.8.2. Data Analysis

To calculate the absorption coefficient and the refractive index of the sample a modified method for extracting the optical constants from terahertz measurements based on the concept introduced by Duvillaret et al. was used [27,28]. The changes in dynamics of the polymer sample were analyzed by investigating the change in the absorption coefficient at a frequency of 1 THz as a function of temperature using the methodology introduced in [21].

## 3. Results

### 3.1. Differential Scanning Calorimetry Data

The calorimetric $T_{g,\mathrm{DSC}}$ was determined for each sample and the resulting values are listed in Table 1. We observed no significant difference in $T_{g}$ between samples. Additionally, we observed one $T_{g}$ for each material, indicating that no phase separation occurred for these samples.

### 3.2. Morphology Measurement Analysis

For each material we report the number of particles counted and the circular equivalent (CE) diameter D[n,0.1], D[n,0.5] and D[n,0.9] percentiles. As shown from Table 2, preparing the microspheres using an oil-in-water emulsification yields products with a relatively broad size distribution for both the blank microspheres and the exendin-4 loaded microspheres before lyophilization. Following lyophilization, the particle size distribution is significantly narrower than before. This could be due to the removal water and moisture [5,9]. The Karl Fischer measurements we conducted revealed that the residual moisture for each vial was less than 1%. Notably, upon lyophilization, as water molecules are removed peptides and copolymers, for example, exendin-4, and PLGA microspheres, can form an extensive hydrogen bonding network [11,29,30] leaving few sites for bonding with water molecules. Additionally, the system is exposed to various stresses of temperature and pressure, and these could cause two or more adjacent pores to merge, as common pore walls rupture, and adjacent particles can combine. Thus, the removal of water in the drying step of lyophilization would affect the size distribution of the dry particles produced, and we can obtain a more narrow particle size distribution compared to a sample in solution [31]. It is worth noting that exendin-4 is inherently flexible and thus, chemical stability is of primary concern; however, given that exendin-4 is a peptide, it lacks a defined protein domain with a characteristic architecture that can undergo unfolding upon exposure to acute freezing and dehydration stresses of lyophilization [32,33]. Finally, when comparing the CE values of the lyophilized samples, the blank PLGA 50:50 samples have a higher CE compared to the 50:50 loaded samples, while the blank 75:25 samples have a lower CE compared to the 75:25 loaded samples. This can be explained by considering emulsion stability which dictates microsphere size. For example, exendin-4 would change the emulsion stability through surfactant-like activity, and the viscosity of the emulsion can increase with higher polymer concentration, polymer MW, and hydrophobicity [34]. Specifically, as the polymer is more hydrophobic, as is the case for PLGA 75:25 due to its a higher lactide fraction, more energy would be required to generate smaller droplets; while for PLGA 50:50 it is expected that smaller droplets could be generated with less energy input; however these parameters were not optimized for and beyond the scope of this work [33,34].

### 3.3. Helium Pycnometry Analysis

The pycnometry data shows that increasing exendin-4 loading increases the density of the material (Table 3). Our helium pycnometry measurements agree with values reported in the literature [4].

### 3.4. Scanning Electron Microscopy (SEM) Characterization

Using SEM, qualitative analysis of a representative set of images indicated that the microspheres examined are predominantly spherical in shape and exhibit a smooth and porous surface (Figure 1, see Appendix A). The internal structure of fractured spheres revealed a porous interior. The qualitative analysis of the data showed that for the blank PLGA microspheres, there is no significant difference visually in the porosity of the blank 50:50 and 75:25 systems, respectively. In contrast, for both the 50:50 and 75:25 systems, the microspheres with high loading of peptide appeared more porous compared to the microspheres which contained a low loading of peptide. See Appendix A for SEM data for blank PLGA 50:50 and 75:25 microspheres, and low polypeptide loaded PLGA 50:50 and 75:25 microspheres.

### 3.5. Fourier Transform Infrared Spectroscopy

FTIR was performed for the PLGA 75:25 microsphere samples (Figure 2). The absorbance of the peaks at around 3300cm${}^{-1}$ and 2950cm${}^{-1}$ increased in intensity with an increase in polypeptide loading. The peak at 3300cm${}^{-1}$ originates from the -OH stretch in exendin-4, and the peak at 2950cm${}^{-1}$ is assigned to the C-H stretching modes in exendin-4 [26,35]. The amide I band at 1600–1700 cm${}^{-1}$ can be attributed to the –C=O stretch, with contributions from the out-of-phase –CN stretch, –CCN deformation, and –NH in-plane bend and is sensitive to the structure of the protein backbone. Additionally, there is an increase in absorption in the amide I region around 1600–1700 cm${}^{-1}$, and a shift in the frequency of the mode for the sample with a high polypeptide loading, which is indicative of changes in the hydrogen bonding network [22]. Thus, the FTIR spectra confirm the increase in exendin-4 loading between the blank, low and high polypeptide formulations, and, with this, a change in the hydrogen bonding network. Notably, it was not possible to investigate an only exendin-4 control, as a stable product could not be freeze-dried.

### 3.6. Analysis of THz-TDS Data

The terahertz spectra of all the microspheres showed an increase in absorption with frequency and temperature over the entire investigated range in line with previous measurements of amorphous molecular solids (Figure 3). As expected for non-crystalline materials, no discrete spectral features were present and the spectra were dominated by the monotonous increase with frequency that is characteristic for the rising flank of the peak due to the vibrational density of states (VDOS) [24]. In contrast, the refractive index subtly decreases with increasing frequency. To further investigate the relationship between the increase of absorption coefficient and temperature we examined the temperature-dependent changes in absorption losses at a frequency 1THz in more detail. Given the lack of distinct spectral features we chose the frequency of 1THz. The rationale for this choice is based on the fact that the signal-to-noise ratio of the measurement at this frequency is high and that we know from our previous work that at a frequency of 1THz the minimum in losses in the spectral response is exceeded for all samples studied and hence the absorption is clearly dominated by the VDOS [27].

The changes in absorption at a frequency of 1THz with temperature for the microsphere samples of PLGA 50:50 and PLGA 75:25, are plotted in Figure 4 and Figure 5 respectively. For all the materials the absorption coefficient was found to increase in a linear fashion with increasing temperature and several distinct temperature regions can be identified for each material. $T_{g,\beta }$ was defined as the intersection point of the two best-fit linear fits at low temperatures for all cases, and $T_{g,\alpha }$ was defined as the intersection point of the two best-fit linear lines at high temperatures, as outlined in [21].

For all of the microsphere samples the change in absorption with temperature can be observed to take place over three distinct regions and two transition temperatures, $T_{g\beta }$ and $T_{g\alpha }$, as determined using the methodology outlined above (see Figure 4 and Figure 5 and Table 1). In this work we attempt to explain the origin of the transition temperatures by proposing a physical picture of the change in the microsphere dynamics with temperature and relate this to the free volume approach, and to peptide and polymer interactions. It is worth noting that the values of $T_{g,\alpha }$, as determined from the THz-TDS experiments, are in good agreement with our own calorimetric measurements, $T_{g,\mathrm{DSC}}$, as well as the values reported in the literature for these materials [36].

## 4. Discussion

### 4.1. Understanding Peptide and Copolymer Interactions

It has been shown that the α-relaxation process is associated with large-scale mobility, whereas the secondary or β-relaxation process is thought to be associated with local mobility, or small-scale mobility [24,37]. To date, there remains a discussion in the literature as to the origin and molecular mechanisms associated with the α- and β-relaxation processes. One secondary relaxation, the Johari-Goldstein (JG) β-relaxation, also referred to as the slow β-relaxation, is considered a universal feature of all amorphous materials [38,39]. This process is observed at higher frequencies than the α-relaxation, and has been associated predominantly with the intermolecular degrees of freedom of a material [16,38,40]. Notably, recent experimental and theoretical work clearly highlights that the potential energy surface (PES) model proposed by Goldstein almost half a century ago is the most intuitive and comprehensive model to understand the molecular dynamics in amorphous systems and that intra- and intermolecular processes are always fundamentally coupled by means of the PES [41,42]. Goldstein explained that as a liquid flows it can move on the PES from one minimum to another minimum, and each minimum is associated with an energy barrier, yet as the liquid moves the volume nor energy of the liquid changes. When the liquid is cooled down to a glass, the liquid structure is trapped in a deep minimum, with some level of mobility remaining as a distribution of relaxation times [42]. Thus, the work of Goldstein provides an intuitive illustration that a liquid can exist in numerous transient structures and it is the potential energy barriers which determine the molecular motions of the viscous liquid which forms a glass: a picture which can be applied to bulk amorphous systems.

The different molecular motions of a copolymer chain can be tracked with changes in temperature. At low temperatures, the copolymer chains are completely disordered and are densely packed, and the motions of the copolymer are restricted. Upon heating to $T_{g,\beta }$, the activation energy and free volume is sufficient for the copolymer to undergo local motion, giving rise to the β-relaxation processes [40]. Previously, we have shown that the $T_{g,\beta }$ is fundamentally linked to the onset of motions in an organic molecule that results in changes of the dihedral angle of one or multiple bonds in the system. Specifically, for PLGA, this could involve local motions of small segments of the copolymer backbone and side chain groups [37,43]. With a further increase in temperature, the copolymer chains are more loosely packed and the activation energy and free volume of the system increases further, allowing for intermolecular large-scale copolymer motions to occur at the temperature indicated by $T_{g,\alpha }$ associated with the α-relaxation process.

Exendin-4 is a polypeptide composed of 39 amino acids, and its structure is thought to resemble a random coil chain [7,44]. The short-range local interactions of a polypeptide influence the conformational preferences of its amino acid chain [45,46,47]. In general terms, it is well established that the partial double bond character of a peptide bond gives rise to its planar structure, and free rotation is restricted about this bond [47]. We hypothesize that with sufficient thermal energy and free volume present in the system, two local motions could take place for exendin-4, which would give rise to the β-relaxation process: (1) local side chain rotations, and (2) local rotation about two single bonds [45,48]. Given that the amino acids of the exendin-4 polypeptide chain are linked by peptide bonds, rotational freedom arises from the single bonds between an amino group and the α-carbon atom and the carbonyl group of the peptide backbone [47].

However, the rotation of these bonds is limited by steric hindrance [47,48]. Based on the chemical structure of the exendin-4 this polypeptide can act as a hydrogen bond acceptor via its carbonyl groups (–C=O) or as a hydrogen bond donor via its amine group (–NH), and its hydroxyl groups (–OH) can act as both the hydrogen bond donor or acceptor. The linear PLGA—${\left[C_{3}H_{4}O_{2}\right]}_{x}{\left[C_{2}H_{2}O_{2}\right]}_{y}$—chains include the methyl side groups of poly(lactic acid) (PLA) and oxygen atoms at every third position of the copolymer backbone, and C=O bonds which introduce significant structural rigidity to the copolymer backbone. Specifically, hydrogen bonds can form: (1) between the carbonyl groups of PLGA and the amine groups of exendin-4, (2) between the amine groups of exendin-4 and the hydroxyl group of PLGA, and (3) between the hydroxyl groups and the carbonyl groups of exendin-4 and PLGA. Additionally, van der Waals as well as dipole-dipole interactions between the peptide groups and the hydroxyl groups can serve to stabilize the system [49]. These strong interactions, the dynamics of which are infrared active, can reduce molecular mobility and improve the physical stability of these systems [3,29,30]. Notably, it is conceivable that with additional free volume and thermal energy input large-scale rotational motions of the backbone dihedral angles in the polypeptide chain could occur, which could lead to large-scale changes in chain conformation, contributing to the α-relaxation.

### 4.2. Tracking the Dynamics of Microspheres Using THz-TDS

We observe three regions with two distinct transition points, $T_{g,\beta }$ and $T_{g,\alpha }$ for both the PLGA 50:50 and PLGA 75:25 microspheres (Figure 4). For the PLGA 50:50 formulation, the high exendin-4 loaded microspheres have a significantly higher $T_{g,\beta }=219\;$K, than the blank and low polypeptide loaded microspheres ($T_{g,\beta }=167\;$K, and $T_{g,\beta }=168\;$K, respectively). In the polymer dispersion, exendin-4 can form hydrogen bonds with PLGA. Intuitively, an increase in exendin-4 loading would correspondingly increase the hydrogen bonding interactions between the polypeptide and PLGA [48], reduce the molecular mobility of the system, which would raise the value of $T_{g,\beta }$ [9,50]. Indeed, we observe a significant increase in $T_{g,\beta }$ for the high polypeptide loaded microspheres, compared to the $T_{g,\beta }$ value observed for the blank and low polypeptide loaded microspheres. Notably, for the blank and low exendin-4 loaded 50:50 microspheres, the onset for local mobility occurs at approximately the same value of $T_{g,\beta }$. This could suggest that the low polypeptide loaded microsphere behave similarly to the blank microspheres, due to the limited interaction between peptide and polymer [46,51]. For the blank microspheres and low exendin-4 loaded microspheres, the onset for local mobility occurs at approximately the same value of $T_{g,\beta }$, ($T_{g,\beta }=167\;$K). For the high exendin-4 loaded microspheres, $T_{g,\beta }$ increases significantly ($T_{g,\beta }=219\;$K). As more polypeptide is added to the PLGA matrix it forms an extensive hydrogen bonding network, which in turn reduces the configurational entropy of the system [44,45,51]. The resultant interactions between exendin-4 and PLGA appear to be stronger compared with the interactions between adjacent PLGA chains.

For the samples of the PLGA 75:25 microspheres, we observe a similar trend to the PLGA 50:50 microspheres. The value of $T_{g,\beta }$ increases from 179K for the blank microspheres, increases further to 192K for the low exendin-4 loaded microspheres, and rises to $T_{g,\beta }=215\;$K, for the high exendin-4 loaded microspheres (Figure 5). Notably, we have previously shown that the methyl side group of PLGA 75:25 inhibits polymer mobility and introduces steric hinderance [43]. Thus, for PLGA 75:25, the steric hinderance caused by the lactide monomer can restrict the intermolecular interactions between exendin-4 and PLGA, limiting the sites that are able to participate in hydrogen bonding, most likely making the carbonyl group of PLGA more favorable for hydrogen bonding. With limited sites for hydrogen bonding, less hydrogen bonds can form between exendin-4 and PLGA, resulting in PLGA chain entanglement, lower mobility, and reduced free volume. Our measurement that the PLGA 75:25 microspheres exhibit higher $T_{g,\beta }$ values compared to the 50:50 microspheres could therefore be explained by steric effects.

Finally, we observe that for both PLGA 50:50 and PLGA 75:25 the value of $T_{g,\alpha }$ increases in the order of blank < low polypeptide loaded < high polypeptide loaded microspheres (Figure 5). This suggests that at high temperature, sufficient activation energy and free volume must be available to facilitate mobility of the polymer and polypeptide. Thus, with increase in exendin-4 loading, due to steric hinderance the threshold for mobility increases, as reflected in the raised value of $T_{g,\alpha }$ (Figure 6). Ideally if it were possible to produce a stable freeze-dried exendin-4 product, it may be feasible to determine whether the exendin alone could aggregate and lead to the changes observed.

## 5. Conclusions

We have produced and characterized lyophilized blank PLGA microspheres and lyophilized PLGA microspheres containing two different loadings of the polypeptide exendin-4. We studied the dynamics and relaxations and the glass transition behavior of these PLGA microspheres by performing variable temperature THz-TDS measurements. A monotonous increase of absorption coefficient with temperature was observed for all the materials examined, and all of the microspheres exhibit three temperature regimes, with a distinct $T_{g,\beta }$ and $T_{g,\alpha }$. We explain our experimental results using the concepts of free volume and discuss the interactions of the polypeptide and copolymer matrix and steric effects. We define $T_{g,\beta }$ as the point at which the material has sufficient amount of activation energy and free volume to allow for local motions to occur, and $T_{g,\alpha }$ as the point at which large-scale movement can take place, and relate the onset of the transition temperatures to the interaction strength between the polymer and the peptide. Our work provides a physical explanation for the behavior of these microspheres leading to $T_{g}$, and agrees with the PES concept outlined by Goldstein [42]. This work provides a framework for understanding the dynamics of complex systems, such as lyophilized microspheres, and considers the parameter of $T_{g,\beta }$ as a valuable criterion for preparing stable formulations. Finally, this work demonstrates that THz-TDS is an effective method to measure the molecular dynamics and temperature-dependent behavior of a polymer-polypeptide microsphere system.

## Figures and Tables

**Figure 1 pharmaceutics-11-00291-f001:**
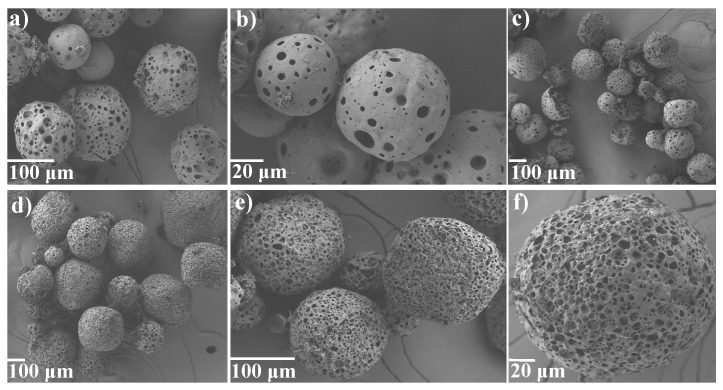
Representative SEM micrographs for low exendin-4 loaded PLGA 75:25 microspheres shown in (**a**–**c**), and high exendin-4 loaded PLGA 75:25 microspheres shown in (**d**–**f**).

**Figure 2 pharmaceutics-11-00291-f002:**
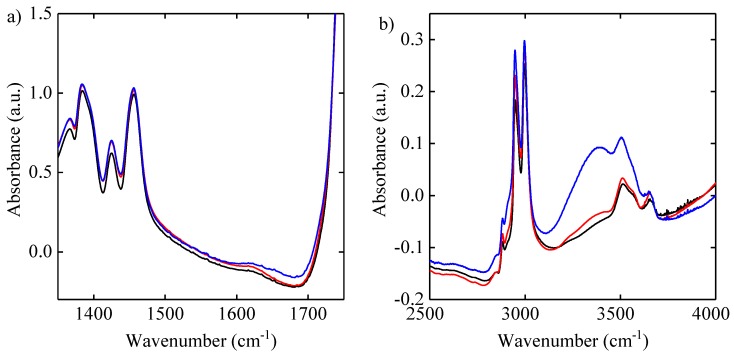
FTIR spectra of blank microspheres (solid black line), low polypeptide loaded (solid red line) and high polypeptide loaded (solid blue line) PLGA 75:25 microspheres. (**a**) shows the wavenumber range of 1300−1700cm${}^{-1}$ and (**b**) shows the wavenumber range of 2500–4000 cm${}^{-1}$.

**Figure 3 pharmaceutics-11-00291-f003:**
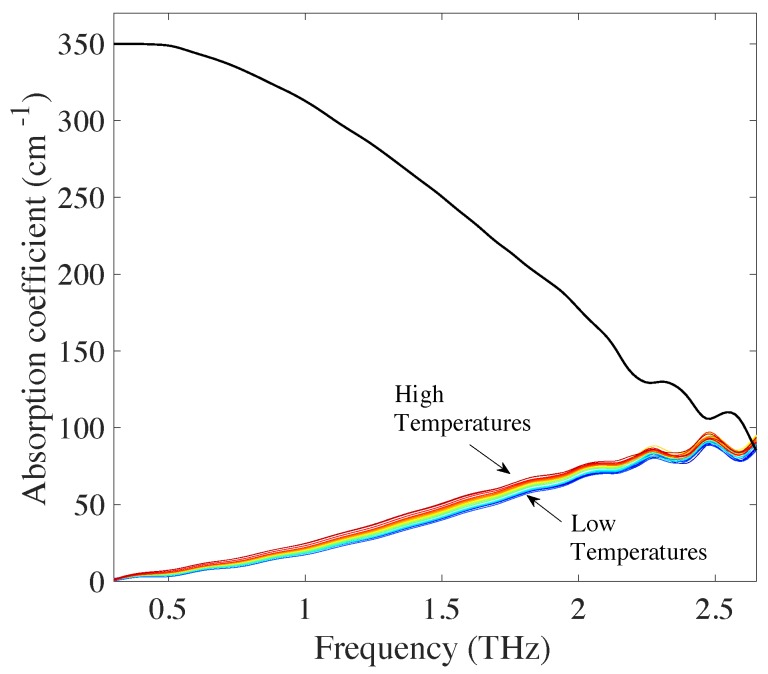
Absorption coefficient spectra of high MW PLGA 75:25 in the temperature range of 100–350 K, with 10K temperature increments between spectra. Solid black line indicates the maximum absorption coefficient.

**Figure 4 pharmaceutics-11-00291-f004:**
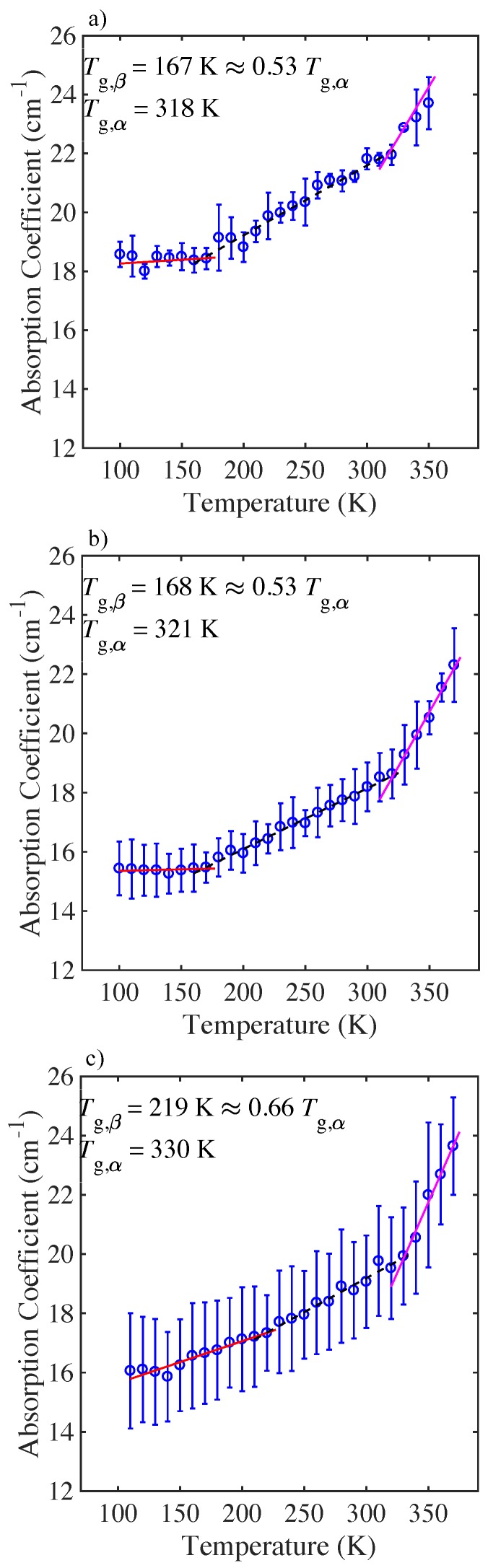
Mean terahertz absorption coefficient as a function of temperature at 1THz for PLGA 50:50 microspheres. Error bars represent the standard deviation for *n* samples. (**a**) blank (n=3), (**b**) low (n=4), and (**c**) high peptide loading (n=3). Lines show the different linear fits of the respective regions.

**Figure 5 pharmaceutics-11-00291-f005:**
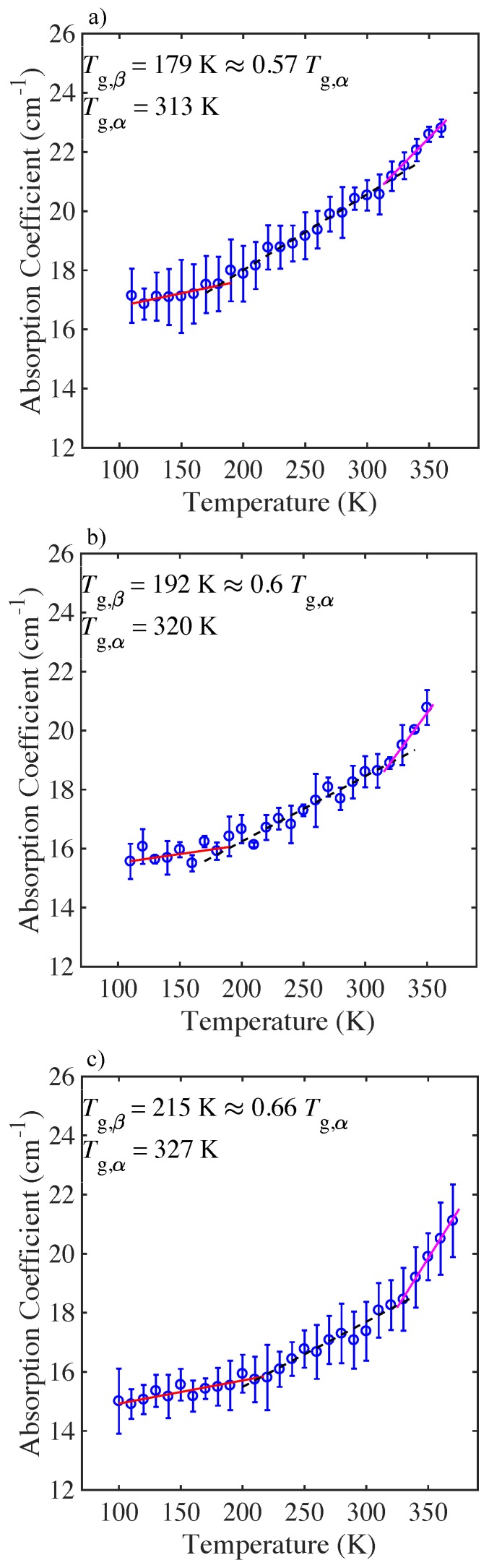
Mean terahertz absorption coefficient as a function of temperature at 1 THz for PLGA 75:25 microspheres. Error bars represent the standard deviation for *n* samples (**a**) blank (n=3), (**b**) low (n=3), and (**c**) high peptide loading (n=3). Lines show the different linear fits for the different regions.

**Figure 6 pharmaceutics-11-00291-f006:**
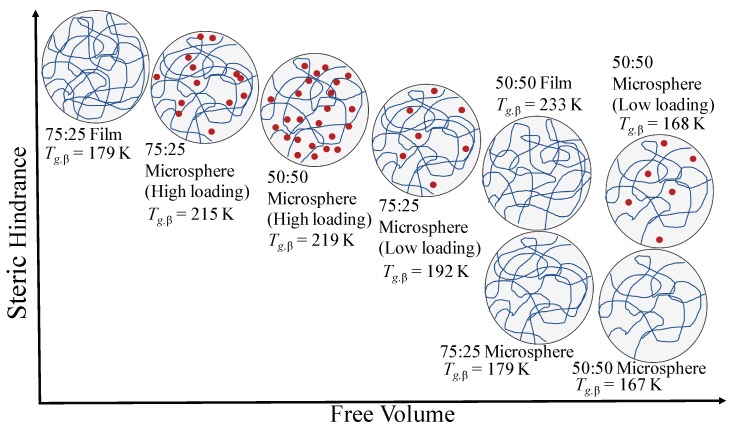
Behavior of the different materials with increasing steric hinderance and free volume. Blue lines represent the PLGA copolymer (20–30 kDa MW), and solid red circles represent the exendin-4 molecules (4.2 kDa MW). Low loading and high loading refers to 1 mg/mL and 10 mg/mL of polypeptide loaded in PLGA microspheres, respectively. With an increase in free volume and decrease in steric hinderance, the values of $T_{g,\beta }$ is decreased.

**Table 1 pharmaceutics-11-00291-t001:** Gradient, *m*, of the linear fit (y=mx+c) for the respective temperature regions as outlined in Section 2.8.2 as well as the respective glass transition temperatures determined based on the terahertz analysis and by DSC.

Material	Peptide Loading in Aqueous Phase (% m/v)	Region 1 (cm${}^{-1}\;$K${}^{-1}$)	Region 2 (cm${}^{-1}\;$K${}^{-1}$)	Region 3 (cm${}^{-1}\;$K${}^{-1}$)	$T_{g\beta }$ (K)	$T_{g\alpha }$ (K)	$T_{g,\mathrm{DSC}}$ (K)
PLGA 50:50	0 (blank)	0.0026±0.0040	0.0237±0.0010	0.070±0.018	167	318	318
PLGA 50:50	1 (low)	0.0010±0.0012	0.021±0.00062	0.074±0.0044	168	320	316
PLGA 50:50	10 (high)	0.014±0.0013	0.023±0.0012	0.095±0.0052	219	330	317
PLGA 75:25	0 (blank)	0.0087±0.0022	0.026±0.00095	0.043±0.0043	179	313	317
PLGA 75:25	1 (low)	0.0060±0.0016	0.022±0.0014	0.057±0.0024	192	320	320
PLGA 75:25	10 (high)	0.0078±0.0015	0.022±0.0013	0.067±0.0021	215	327	322

**Table 2 pharmaceutics-11-00291-t002:** Morphology data for unlyophilized samples and lyophilized samples. CE is the circular equivalent diameter: the diameter of a circle with the same area as the 2D projection image of the particle.

Ratio	Lyophilized	Peptide Loading in Aqueous Phase (% m/v)	Particles Counted (Number)	CE D[n,0.1] (μm)	CE D[n,0.5] (μm)	CE D[n,0.9] (μm)
50:50	No	0	3218	77	119	176
50:50	No	1	1108	45	76	110
50:50	No	10	1108	45	76	110
50:50	Yes	0	763	73	100	136
50:50	Yes	1	2321	75	116	159
50:50	Yes	10	3839	68	110	156
75:25	No	0	1909	55	106	167
75:25	No	1	2479	68	114	170
75:25	No	10	361	111	178	242
75:25	Yes	0	8465	63	112	163
75:25	Yes	1	903	94	154	197
75:25	Yes	10	1477	82	147	186

**Table 3 pharmaceutics-11-00291-t003:** Helium pycnometry data for lyophilized microsphere samples and PLGA 50:50 and 75:25 copolymer films.

Material	Peptide Loading in Aqueous Phase (% m/v)	Density (g/cm${}^{3}$)
PLGA 50:50 Film	0	1.39
Microsphere PLGA 50:50 blank	0	0.97
Microsphere PLGA 50:50	1	1.66
Microsphere PLGA 50:50	10	2.09
PLGA 75:25 Film	0	1.36
Microsphere PLGA 75:25 blank	0	1.41
PLGA 75:25	1	1.56
PLGA 75:25	10	1.62

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
