# Peer review of "Terahertz Spectroscopy: An Investigation of the Structural Dynamics of Freeze-Dried Poly Lactic-co-glycolic Acid Microspheres"

_pharmaceutics, 2019, doi:10.3390/pharmaceutics11060291_

Round 1

Reviewer 1 Report

The authors investigated the structural dynamics of freeze-dried PLGA microspheres with and without the peptide (extendin-4) using temperature-variable terahertz time-domain spectroscopy (THz-TDS). The molecular dynamics and relaxations and the glass transition behaviors of PLGA microspheres were investigated. The approaches using THz-TDS are interesting and considered to be new to this field. For this reason, the paper can be published after minor changes to address the issues in below.

- Please add the molecular weight of PVA.

- Peptide loading values in Tables 1, 2 and 3 are not correct as the values are the amount of the peptide which were simply added to the aqueous phase.

- Please add any experimental data showing the extendin-4 was intact after being loaded in PLGA microparticles.

Reviewer 2 Report

The manuscript titled “Terahertz Spectroscopy: An investigation of the Structural Dynamics of Freeze-Dried PLGA Microspheres” is investigating the dynamics of PLGA microspheres prepared by freeze-drying and the molecular mobility at lower temperatures leading to the glass transition temperature, using temperature-variable terahertz time-domain spectroscopy (THz-TDS) experiments. The manuscript is well written and the methodology section is well described. The manuscript presents some interesting data, yet there are some concerns on how the experiment were designed and this might affect the results making it inconclusive. For instance, the stability of the freeze dried peptide was not assessed.

Below are more specific comments for the authors to consider.

Methodology

4.2. Specify how many times were the microparticles washed?

4.3. Helium Pycnometry Measurement

Provide details on any prep-treatment conditions for these samples prior to the measurement using the helium pycnometer.

Move section 4.6. Sample Preparation under Terahertz Time-Domain Spectroscopy (THz-TDS)

Results and discussion

Table 2, it is quite interesting that the non-lypholised samples are smaller in size compared to the lypholised ones. Also for PLGA 75:25 why the non-loaded samples are bigger in size. It is the other way round for PLGA 50:50. The authors need to look at this more closely and provide a justifiable explanation.

FTIR was used to determine the secondary structure of PLGA microsphere, yet the preparation method is aggressive and could change the structure of the polymer and peptide. ATR would have been more appropriate.

SEM, the authors did not comment on the surface roughness of the lypholized samples and samples total porosity?  Also the pore size and structures of lyophilized samples are hugely affected by the freezing step, can authors provide details how samples were frozen (is it flash freezing, slow freezing, over how many minutes and what are the freezing temperatures…etc) prior to freeze drying.

Acute freezing and dehydration stresses of lyophilization can induce protein unfolding, have the authors assessed the stability of exendin-4 and how unfolding could affect the results.

Presentation

The authors are using “We” very often and it is better to use third narrator all the time.

Table 2. Morphology data for unlyophilised samples and lyophilised samples indicated by an X symbol. This is confusing probably use a dash for non-lyophilized and tick for lyophilised or yes/no?

Table 1 and table 2 are not cited in text before their appearance.

[22? Remove the question mark

Author Response

See attached comments

Reviewer 3 Report

The authors attempt to provide THzTDS  analysis of PLGA microspheres. Unfortunately the manuscript lacks of any discussion on the retrieval of PLGA electrodynamics. Authors should apply a mean field approch (Maxwell-garnett, Landau...) to extract PLGS electrodynamics from KBr matrix. It would be recommended to use in the paper both real and immaginary parts of retrieved parameters because looking to your data is clear that both contain fruitful information.  Furthermore, the way Authors state the aim of their research is rather confusing because in either the introduction and result sections the "paper target" is mentioned three different times (lines 33,72,137). 
Authors on one side should spend more time to provide a clear message of their scientific target, and on the other the THz analysis should be dealt to furnish the minimal information to reproduce either experiments and results.     

Author Response

Seee attached comments

Reviewer 4 Report

The manuscript is original and interesting. The experimental data are transparent and logically justified, and the conclusions are outlined properly. Therefore I recommend the acceptance of this manuscript in its present form.

Author Response

No comments to address from Referee 4.

Round 2

Reviewer 2 Report

I would like to thank the authors for addressing my comments. I am happy with the manuscript and looking forward to seeing it published online.

Reviewer 3 Report

I feel to say that manuscript is now ready to be published